# Inferring the age and environmental characteristics of fossil sites using citizen science

Tara Djokic[1,2,3]*, Michael Frese[1,4,5], Adam Woods[1], Mary Dettmann[1,6], Paul Flemons[1], Frank Brink[7], Matthew R. McCurry[1,2,8]

1 Australian Museum Research Institute, Sydney, New South Wales, Australia, 2 ESSRC, School of Biological, Earth and Environmental Sciences (BEES), University of New South Wales, Kensington, New South Wales, Australia, 3 School of Earth and Environmental Sciences, The University of Queensland, St Lucia, Queensland, Australia, 4 Commonwealth Scientific and Industrial Research Organisation, Health and Biosecurity, Black Mountain, Australian Capital Territory, Canberra, Australia, 5 Faculty of Science and Technology, University of Canberra, Bruce, Australian Capital Territory, Australia, 6 Geosciences, Queensland Museum, Queensland, Australia, 7 Centre for Advanced Microscopy, Australian National University, Acton, Australian Capital Territory, Australia, 8 Paleobiology, National Museum of Natural History, Smithsonian Institution, Washington D.C., United States of America

* t.djokic@unsw.edu.au

**Data Availability Statement:** All raw expedition data (CSV files) and collated data has been included in the Supporting information file.

**Funding:** M.R.M. and M.F. received funding from: 1. Australian Museum Foundation (no grant

## Abstract

Not all fossil sites preserve microfossils that can be extracted using acid digestion, which may leave knowledge gaps regarding a site's age or environmental characteristics. Here we report on a citizen science approach that was developed to identify microfossils in situ on the surface of sedimentary rocks. Samples were collected from McGraths Flat, a recently discovered Miocene rainforest lake deposit located in central New South Wales, Australia. Composed entirely of iron-oxyhydroxide, McGraths Flat rocks cannot be processed using typical microfossil extraction protocols e.g., acid digestion. Instead, scanning electron microscopy (SEM) was used to automatically acquire 25,200 high-resolution images from the surface of three McGraths Flat samples, covering a total area of 1.85 cm$^2$. The images were published on the citizen science portal DigiVol, through which 271 citizen scientists helped to identify 300 pollen and spores. The microfossil information gained in this study is biostratigraphically relevant and can be used to constrain the environmental characteristics of McGraths Flat. Our findings suggest that automated image acquisition coupled with an evaluation by citizen scientists is an effective method of determining the age and environmental characteristics of fossiliferous rocks that cannot be investigated using traditional methods such as acid digestion.

## Introduction

Citizen science is a proven method for processing large-scale biological datasets and conveying scientific approaches to a general audience [1–3]. Thus far, most biology-related citizen science projects have assessed extant biodiversity and assisted in conservation management

number issued) https://australian.museum/get-involved/join/foundation/ 2. The Etheridge Family Descendants (no grant number issued) - a philanthropic contribution No URL available 3. An Australian Research Council (ARC) Linkage Grant (LP210301049) https://www.arc.gov.au/ The funders had no role in study design, data collection and analysis, decision to publish, or preparation of the manuscript.

**Competing interests:** The authors have declared that no competing interests exist.

[4–7]. However, citizen scientists have rarely been engaged in palaeontology research for reasons including that fossils can be fragile and difficult to access [8, 9]. At the time of writing, only one palaeontology project was identified from more than 3000 projects listed on the global citizen science hub, SciStarter (using search terms: 'fossil' and 'palaeontology/paleontology'; https://scistarter.org/). For palaeontology to benefit from large-scale data processing, researchers will need to seek innovative ways of connecting with citizen science.

For this study, we used rocks collected from McGraths Flat, a newly discovered Lagerstätte in central New South Wales, Australia [10]. Interpreted as a Miocene age (11–20 million-years-old) rainforest lake deposit, McGraths Flat contains exceptionally well-preserved compressions/impressions of leaves, flowers and fruiting bodies, aquatic and terrestrial insects, spiders, fishes, and other vertebrates (Fig 1) [10, 11]. In addition to these macrofossils, a diverse suite of microfossils, including pollen, spores, yeast cells, nematodes, and insect wing scales are also well-preserved [10].

Microfossils (e.g., pollen and spores) provide an important resource for determining the age and environmental characteristics of fossil sites. Traditionally, microfossils are extracted from sedimentary rocks using a variety of acid dissolution techniques [12, 13]. Acids are chosen based on their ability to dissolve the rock matrix (e.g., silicate, carbonate), while leaving organic (e.g., sporopollenin) or mineralised (e.g., silica, calcareous) microfossils intact [14]. This method is only effective if the fossils and surrounding rock matrix are compositionally

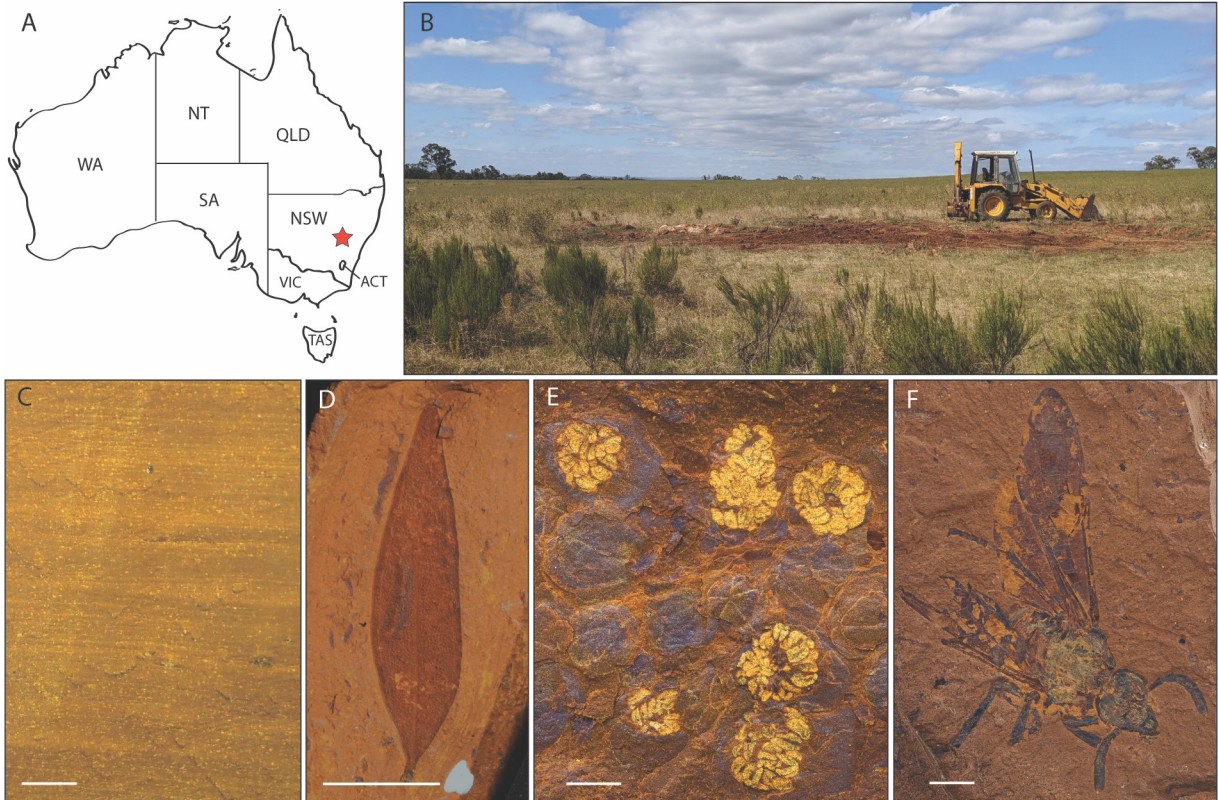

**Fig 1. Location and fossils of McGraths Flat.** (*a*) Map of Australia depicting the location of McGraths Flat (red star) near the town of Gulgong, Central Tablelands, New South Wales, Australia. (*b*) Field site. (*c*) Cross-section of the finely-bedded goethite-rich sedimentary rock from McGraths Flat. (*d*) Myrtaceous (eucalypt) leaf (AM F.146592). (*e*) Budding Malvales(?) flowers (AM F.146589). (*f*) Sawfly (Tenthredinoidea: Symphyta) (AM F.145093). (*d-f*) modified from McCurry *et al.*, 2022. Note the variation in colour (yellow-red-brown) is dominantly due to grain size, the overall composition is homogeneous. Scale bars, *c*—1 cm, *d*—1 cm, *e*—1 mm, *f*—2 mm.

different (heterogeneous) [14]. However, some sites, like McGraths Flat, preserve fossils and rock matrix that are compositionally similar (homogeneous) [10].

At McGraths Flat, fossils and the surrounding sedimentary matrix are almost entirely composed of iron-oxyhydroxide (goethite), which makes it impossible to extract microfossils using traditional strong acid (e.g., HCl, HF) treatments. Likewise, other extraction protocols that rely on relatively weak solutions (e.g., detergent, hydrogen peroxide), disaggregation, or heavy liquid separation [15, 16] are not suitable methods of isolating palynofossils from McGraths Flat. Therefore, in situ observation is presently the only viable method of investigating microfossils in the chemically homogeneous rocks of McGraths Flat.

Using a hammer and chisel, the rocks from McGraths Flat can often be split easily, along bedding planes, which exposes relatively flat surfaces. The high conductivity of the iron-rich rocks and microfossils enables high-resolution (2 nm) SEM imaging without having to apply a conductive coating [10].

There are, however, some disadvantages for in situ microfossil observation compared to traditional extraction methods [13]: (*i*) Operating an SEM is time-consuming, which limits the number of microfossils that can be identified in a reasonable time frame. In 6 hrs of instrument time, a professional scientist can locate and image approximately 50 pollen and spores, which is a relatively small number compared to the hundreds to tens of thousands of specimens that can be acid extracted from as little as five grams of fine-grained clays or 50 grams of sandstone [12]; (*ii*) Rare and relatively featureless types of microfossils may be overlooked due to observation bias (e.g., large pollen with striking features are detected more readily than smaller, less distinguishable specimens); and, (*iii*) Operating an SEM is relatively expensive. Besides the costs associated with the acquisition and maintenance of the microscope, costs include the salary of a qualified specialist, usually a palaeontologist.

These disadvantages prompted us to develop a new observational method for collecting and processing in situ microfossil data. Here, we present a novel citizen science approach that was designed to locate and identify microfossils from tens of thousands of scanning electron microscope (SEM) images taken from the surface of sedimentary rocks. We report on our findings as a case study conducted through the Australian Museum citizen science project 'Date a Fossil'.

## Materials and methods

### Microfossil image acquisition

Three relatively flat, unpolished, and uncoated rock samples (AM F.146572; AM F.152543; AM F.152544) from McGraths Flat were imaged at the Centre for Advanced Microscopy located at the Australian National University, Canberra. Imaging was conducted using an FEI Quanta 650F field emission scanning electron microscope (FE-SEM) that was operated at 15 kV and with 1 nA of probe current (Fig 2*a* and 2*b*). Samples were desiccated for 24 hours prior to automated image acquisition to shorten the times taken to reach the minimum $3 \times 10^{-3}$ Pa Torr required for high vacuum examinations. To enhance the depth of field and account for any surface irregularities in the sample, a relatively small 30 μm objective aperture was used in combination with a relatively long working distance of 15 mm. An automated image acquisition algorithm (FEI MAPS 3.0) was used to produce image montages composed of $60 \times 60$ (3600) SEM images (Fig 2*c* and 2*d*). A total of seven montages yielded an overall dataset of 25,200 SEM images. The acquisition time for each image was set to 8 seconds, resulting in a total time of approximately 8 hours to acquire a 3600-image montage. Each individual image consists of $6144 \times 4090$ pixels, for a 120-μm field of view, limiting the resolution to around 20 nm per pixel. An image overlap of 12.5% was used to facilitate the accurate registration of each

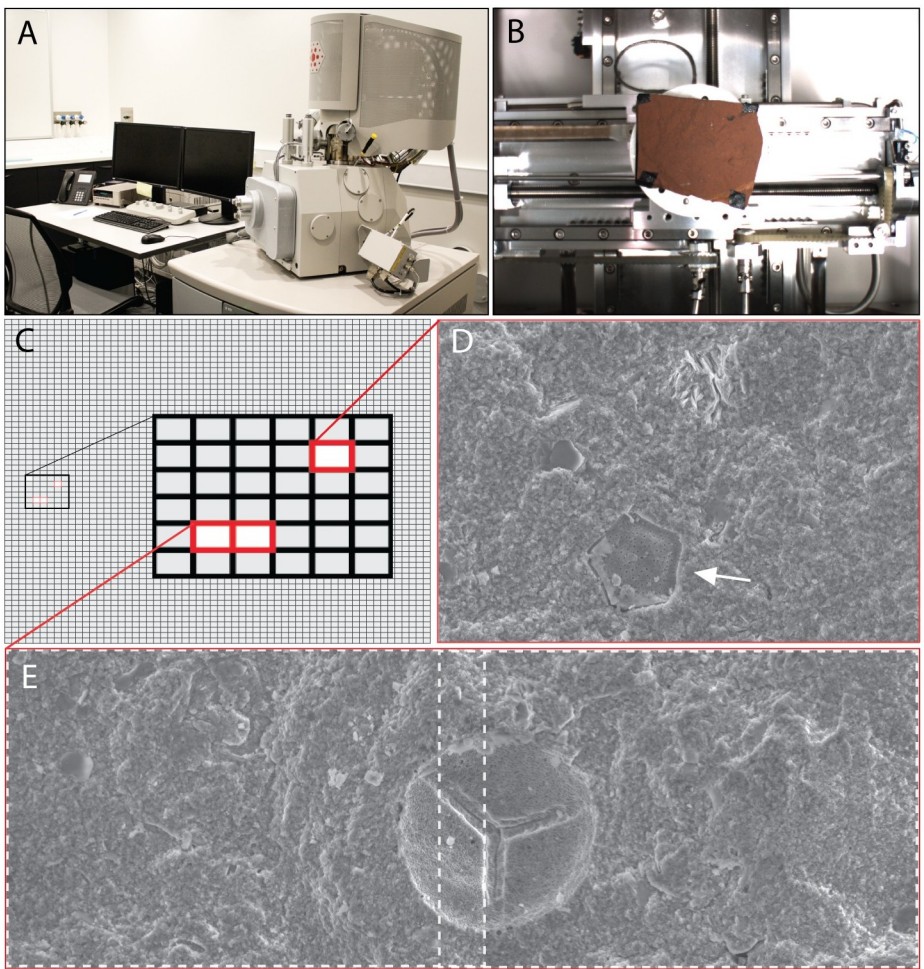

**Fig 2. Automated image acquisition.** (*a*) FEI QEM SCAN 650F scanning electron microscope (SEM). (*b*) A representative rock sample (AM F.147103) from McGraths Flat (approx. $50 \times 60$ mm) used for automated image acquisition; the photograph was taken using the microscope chamber's digital camera. The flat surface of the rocks and their iron-rich composition makes it simple to maintain focus, as well as provide the conductivity needed for SEM operation without applying a conductive coating to the sample. (*c*) Schematic depiction of a $60 \times 60$ (3600) image montage; the area covered by 3600 images is 0.265 cm$^2$ (note that neighbouring images have a 12.5% overlap). (*d,e*) SEM images that contain microfossils: *d*—image showing a single pollen (*Nothofagidites emarcidus/heterus*); and, *e*— two neighbouring images (dashed lines) showing an unknown spore. The images were electronically stitched together using Kolor Autopano Pro 4. The SEM images have not been adjusted for contrast and appear in the same format as viewed by volunteers.

image within the montage (Fig 2e). This resulted in a total imaged area of $6.3 \times 4.2$ mm (0.265 cm$^2$) for each montage. The total area imaged for all seven $60 \times 60$ montages (accounting for overlap) was 1.85 cm$^2$. Each montage was acquired from a different area across the three samples. Collecting images from different areas and samples allowed for a better representation of the microfossil density and diversity, both spatially and temporally.

## DigiVol workflow

DigiVol is an online citizen science platform that was jointly developed by the Australian Museum, the Atlas of Living Australia, and CSIRO (https://volunteer.ala.org.au/). Digivol is used to engage citizen scientists in the digitisation of natural history collections [17]. In this

study, the DigiVol platform was adapted to enable citizen scientists to find and identify micro-fossils (referred to as transcribing) from high-resolution SEM images in a series of online 'virtual' expeditions. SEM images were uploaded into 28 virtual expeditions on DigiVol under the project title 'Date a Fossil'. Each SEM image montage (3600 images) was divided into four expeditions (three expeditions with 1000 images in each and one expedition with 600 images). This was repeated for each of the seven montages. All images in the expeditions were accompanied by a questionnaire that included illustrations (S1 Fig). For each image, citizen scientists were asked to respond to several questions (they also had the option to skip an image). The questionnaire was designed to guide citizen scientists through a 5-step identification process (S1 Fig):

1. Is the image out of focus? (citizen scientists answer by clicking on an image that is either *in focus* or *out of focus*).

2. Are there any microfossils in this image? (citizen scientists answer by clicking on an image *with a microfossil* or *without a microfossil*).

3. How many microfossils are present? (citizen scientists answer by selecting from a drop-down list).

4. Where is the microfossil? (citizen scientists answer by clicking on an image with a microfossil or microfossils positioned in the *middle*, *edge*, or both *middle and edge*).

5. What type of microfossil is present? (citizen scientists answer by clicking on one image from a selection of pollen, spores, and other microfossils).

Anyone could register and participate in 'Date a Fossil' expeditions. The DigiVol platform is accessible via the internet and free to use (http://digivol.ala.org.au). Only one expedition was available to citizen scientists at any one time; participation in the next expedition required completion of the previous expedition.

DigiVol has two mechanisms, consensus and expert verification, to maintain quality assurance of captured data. The consensus approach involved an automated comparison, using a decision tree, of inputs made by multiple citizen scientists. Expert verification was achieved through the manual review of validated images by an experienced scientist (i.e., project officer or palynologist). Similar approaches (using both volunteer and expert steps) have previously been used for validating the identifications of animals from camera trap images, which effectively used two layers of quality control for data analysis [18, 19].

We combined a staged approach of both consensus and expert verification because of: (*i*) time constraints; (*ii*) pilot checking effectiveness; and (*iii*) the intention to review data for use in further research/publication. When three citizen scientists had the same responses to the questions set out in the template, images were marked as 'valid'. If three out of four citizen scientists did not have the same response for an image, it was listed as an 'invalid' image and flagged for expert review/verification. The following details the validation and verification process:

• If 'agreement' was reached by a minimum of three citizen scientists on questions set out in the template, the system would automatically validate the image. The two main identification categories include:

a. Agreement, fossil. In this category, all three citizen scientists agreed that at least one microfossil was present in the SEM image.

b. Agreement, no fossil. In this category, all three citizen scientists agreed that no microfossils were present in the SEM image.

- Disputed images. If a fourth citizen scientist identified a microfossil in an image *before* three others had reached 'agreement, no fossil', the disputed image would be flagged in the system. The system would still label the image with 'agreement, no fossil', but expert reviewers (project officers) would be alerted (in the data) to double-check the image for microfossils.

- If 'no agreement' was reached by a minimum of three citizen scientists on questions set out in the template, the system would label the image for expert review. Microfossils that could not be easily identified by project officers were sent to a palynologist.

## Citizen scientist training and support

Citizen scientists were encouraged to read an 'identification guide' (S2 Fig), an instruction manual that explains the questionnaire protocol and assists in the identification of pollen, spores, and other microfossils. The guide includes schematic representations (illustrations and SEM images) of different microfossil categories along with scientific names and size ranges. Although this guide was recommended reading, it was not mandatory for citizen scientist participation. Six weeks after 'Date a Fossil' was launched on DigiVol, an information webinar was offered to citizen scientists. The webinar provided additional information on the project's scientific background and purpose, as well as a step-by-step guide through the identification process. Examples of microfossils and artefacts that had been recently observed by citizen scientists were displayed. Throughout the webinar, citizen scientists had the opportunity to ask questions and provide feedback. During the entire life of the project, citizen scientists were supported on an ad hoc basis through the DigiVol online discussion forum, where they could post questions and seek technical help.

## Ethics statement

The Human Research Ethics Committee of the University of Canberra has decided that project "11906—Inferring the age and environmental characteristics of fossil sites using citizen science" is exempt from review.

## Results

The 'Date a Fossil' project was online for 3.5 months (March 15–July 1, 2021). In that time, 271 citizen scientists participated in 28 online expeditions, analysing 25,200 SEM images. Each expedition, consisting of either 600 or 1000 images, was completed by an average of 28 citizen scientists in approximately 3–4 days. Each image was viewed up to four times resulting in a total of 81,939 interactions. The average time citizen scientists spent viewing and analysing an individual image was 24.9 seconds, and a total of 566 citizen scientist engagement hours were recorded. Engagement varied widely; contributions ranged from citizen scientists analysing a single image to one individual transcribing 14,509 images. Notably, 85% of the data was completed by only 10% of the transcribers. This top 10% of transcribers completed a mean number of 2565 transcriptions each. The other 90% of transcribers completed a mean number of 52 transcriptions each.

To establish the total pollen and spore count, several steps were taken throughout image processing, from volunteer validation to expert review and verification (Fig 3; S1 Table). First, citizen scientists viewed and categorised all 25,200 images, which significantly reduced the dataset to 4192 images (Fig 3; for raw data files see S1 Data). Project officers then reviewed the 4192 images over a total of 9 hours. Finally, any images of pollen or spores that could not be classified by project officers were sent to a palynologist for identification. This process produced the following results.

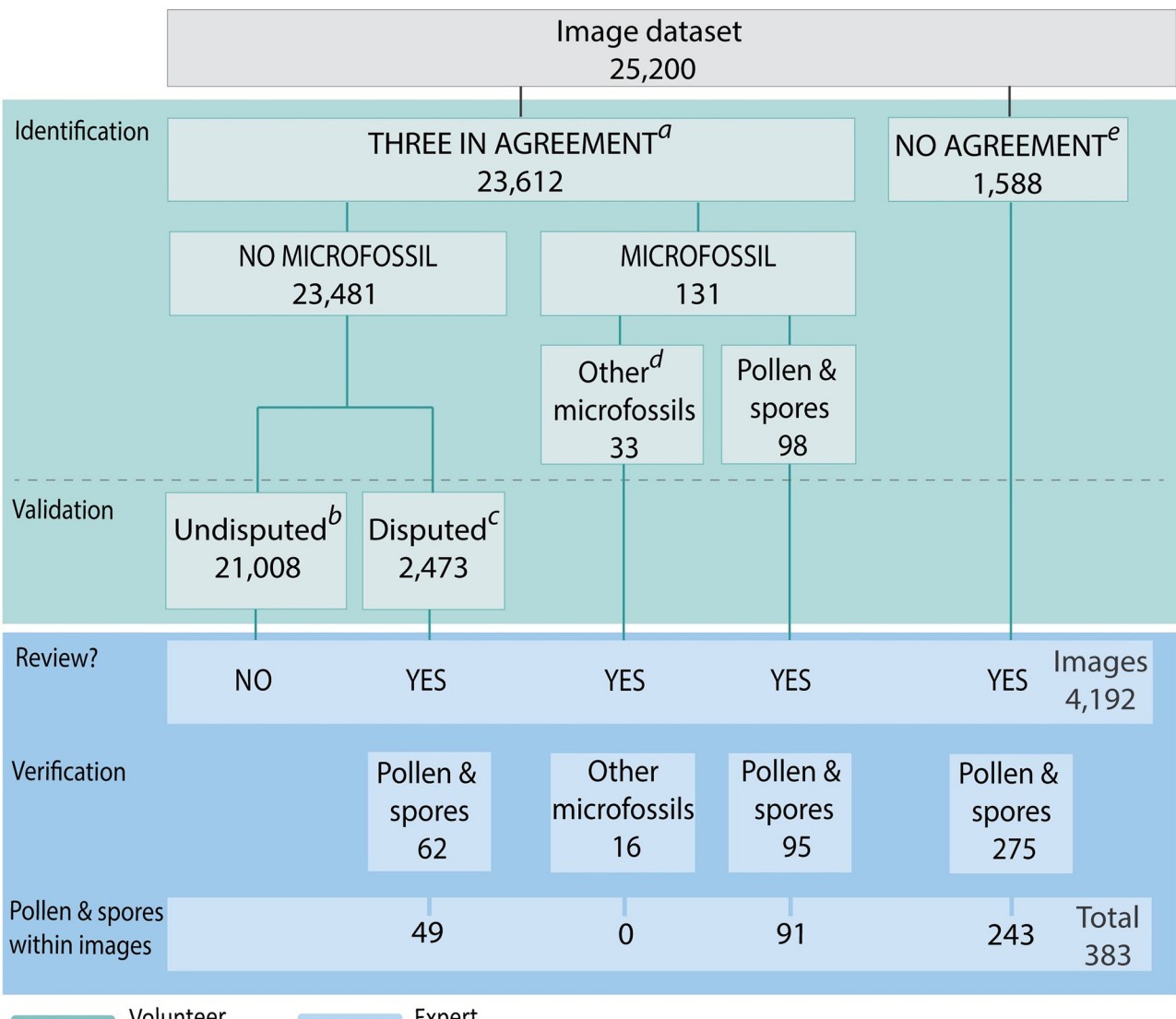

**Fig 3. Flow chart outlining the identification and verification steps involved in the analysis of SEM images.** Citizen scientist (volunteers) microfossil identification reduced the SEM image dataset from 25,200 to 4192. Experts reviewed the 4192 images and verified microfossils in 448 images. Accounting for image overlap, the final pollen and spore count in these verified images was 383, of which 300 specimens were identifiable. Key for superscript lettering: a—An 'agreement' was reached by a minimum of three volunteers on questions set out in the questionnaire template; this category was then subdivided into images indicated as having 'no microfossil/s' and images indicated as having 'microfossil/s'. b—These images were undisputed (i.e., not disputed by a fourth volunteer). c—These images were disputed by an additional (fourth) volunteer and had to be reviewed by an expert. d—These images were identified by three volunteers as containing other microfossils, that are not pollen or spores. e—If 'no agreement' was reached by a minimum of three volunteers on questions set out in the questionnaire template, these images were automatically marked by the system to be reviewed.

An agreement was reached on 23,612 images by three citizen scientists. These images were identified to contain either no microfossil/s (n = 23,481) or microfossil/s (n = 131). From the no microfossil group, 2473 images were disputed by a fourth citizen scientist who indicated they had observed a microfossil. A project officer reviewed the disputed images and found that only 62 images contain pollen or spores (with a total specimen count = 49). From the 131 images identified to contain microfossil/s by three citizen scientists, 98 were labelled with 'pollen or spores', and 33 were labelled 'unknown'. A review of these images by a project officer

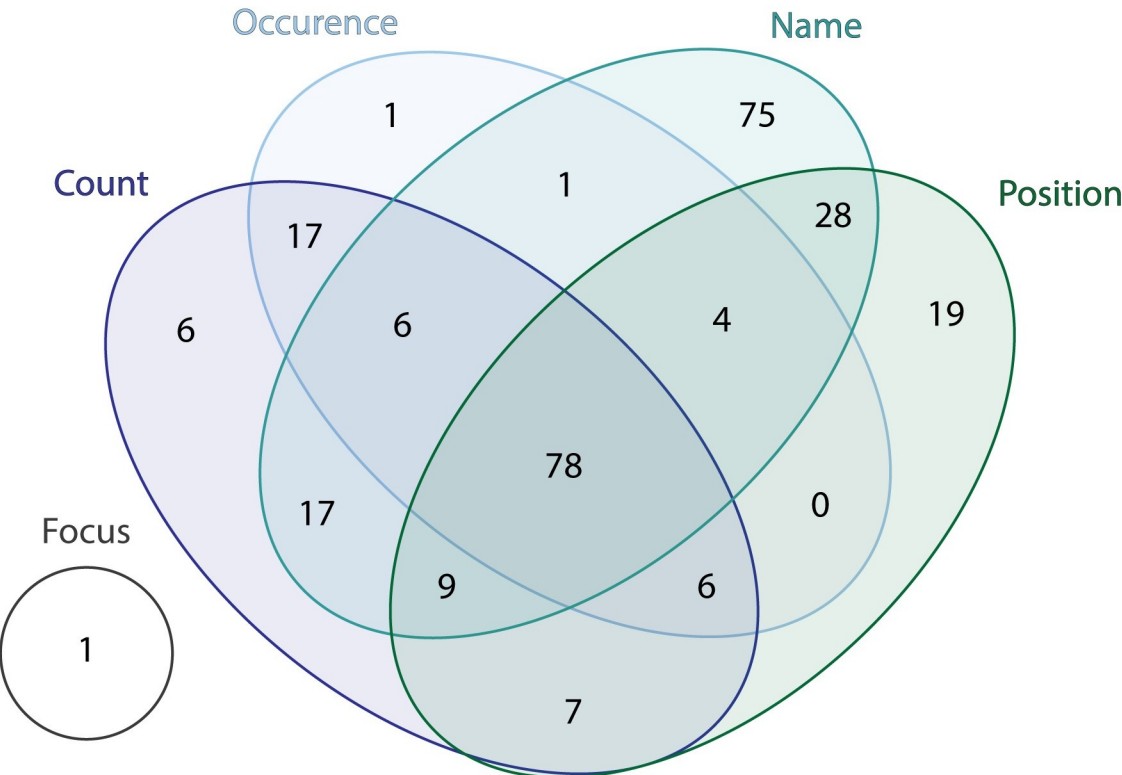

**Fig 4. Reasons for no agreement on images.** Venn diagram illustrating the number of times that questions related to specimen count, occurrence, position, name, and image focus on the questionnaire template led to 'no agreement' among volunteer citizen scientists. The analysis is based on 275 images that were expert verified as containing pollen or spores (see Fig 3 for details). The top reasons that resulted in 'no agreement' included a combination of questions (count, occurrence, position, and name = 78 images), followed by identification of the specimen (75 images). Occurrence—whether or not a microfossil is present in an image; count—number of pollen or spores in an image; position—placement of microfossil on the edge or middle of the image; name—selected from a range of listed specimens; and focus—image is in focus or out of focus.

found that only 95 images contain pollen or spores (with a total specimen count = 91) and only 16 images contain unknown microfossils (Fig 3).

No agreement was reached by three citizen scientists on 1588 images. A review by a project officer revealed pollen or spores in 275 of the 'no agreement' images (with a total specimen count = 243). Citizen scientists did not reach a consensus on the 1588 images for the following reasons: disagreement on the number of microfossils (count); whether microfossils were present (occurrence); the identity (name) of the microfossil; and/or the position of the microfossil within an image (Fig 4). A total of 432 images contain pollen or spores. However, due to image overlap, only 383 specimens are counted. The total pollen and spore count from the entire area analysed (1.85 cm$^2$) includes 300 identifiable specimens and an additional 83 indeterminable specimens (Figs 5 and 6; S2 Table). The top three most abundant palynomorphs among those identified include *Nothofagidites* ssp. (44%; 72.4 specimens/cm$^2$), fungal spores (20%; 32.4 specimens/cm$^2$), and spores of ferns and mosses (9%; 15.7 specimens/cm$^2$). The remaining palynomorphs (27%) contribute significantly fewer population counts i.e., avg 5.9 specimens per palynomorph/cm$^2$ (Fig 5*b*; S2 Table).

To assess whether or not citizen scientists had captured/missed any significant microfossils in the dataset, a project officer manually viewed all 1000 images in Expedition No.1. It was found that seven possible pollen and spores had not been detected by citizen scientists. These

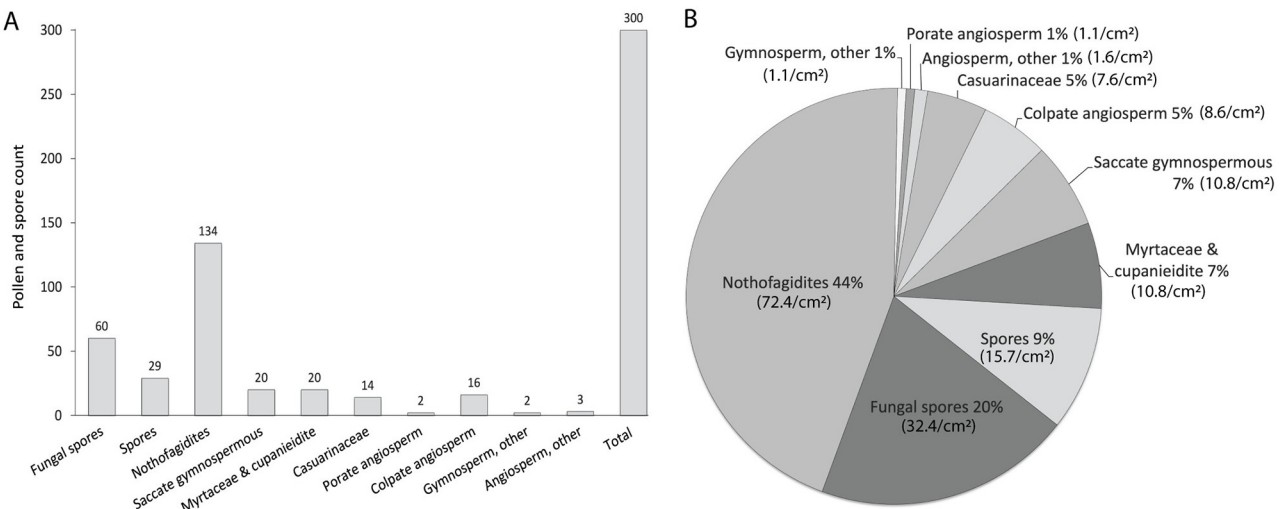

**Fig 5. Microfossil identification, counts, and abundance.** (*a*) The total number of palynomorphs identified across 1.85cm² of rock surface analysed. Note 'Spores' indicates that from ferns and mosses. (*b*) Pie chart showing the relative abundance (%) of palynomorphs, and in brackets, palynomorph density (number of specimens/cm²).

specimens have indeterminable features and therefore attribution to a particular taxon is difficult. From this, we estimate that about 200, likely unidentifiable, pollen and spores could have been overlooked across the entire dataset.

## Discussion

### Time, cost, and research efficiency

Citizen scientists analysed a 25,200-image dataset and identified microfossils or microfossil-like objects in 4192 images. Based on a browsing time of 5 seconds per image, we estimate that a trained project officer could review 25,200 images in approximately 35 hours (one working week). In contrast, a project officer can review the reduced dataset (4192 images) in about 6 hours. The difference is likely to be even greater as these calculations do not take into account the time involved in data organisation and specific identifications (in contrast to recognising a general microfossil occurrence), or accuracy lost to observer fatigue. It is evident that the contribution of citizen scientists can substantially increase both the accuracy and speed at which microfossils are identified in large image datasets.

Automated image acquisition and citizen scientists also reduce SEM costs and enable additional quantitative analyses. Earlier studies of microfossils from McGraths Flat provide some reference points [10]. An experienced professional scientist requires a full working day (about 6 hours of instrument time) to discover and image approximately 40–50 pollen and spores. However, the scientist used a less structured approach that involved scanning the rock surface at different and constantly changing magnifications to cover as much of the rock surface as possible. Furthermore, the scientist often ignored specimens that are only partly preserved, not well presented, or belong to species that have already been imaged. Thus, although professional scientists may be better at collecting high-quality images, the citizen science approach generates much more quantitative data (Fig 5). Furthermore, citizen scientists reduce bias through a more structured, systematic image analysis involving multiple individuals; it may also lead to the detection of smaller, less obvious, or less abundant microfossils (e.g., Fig 6*h* and 6*i*).

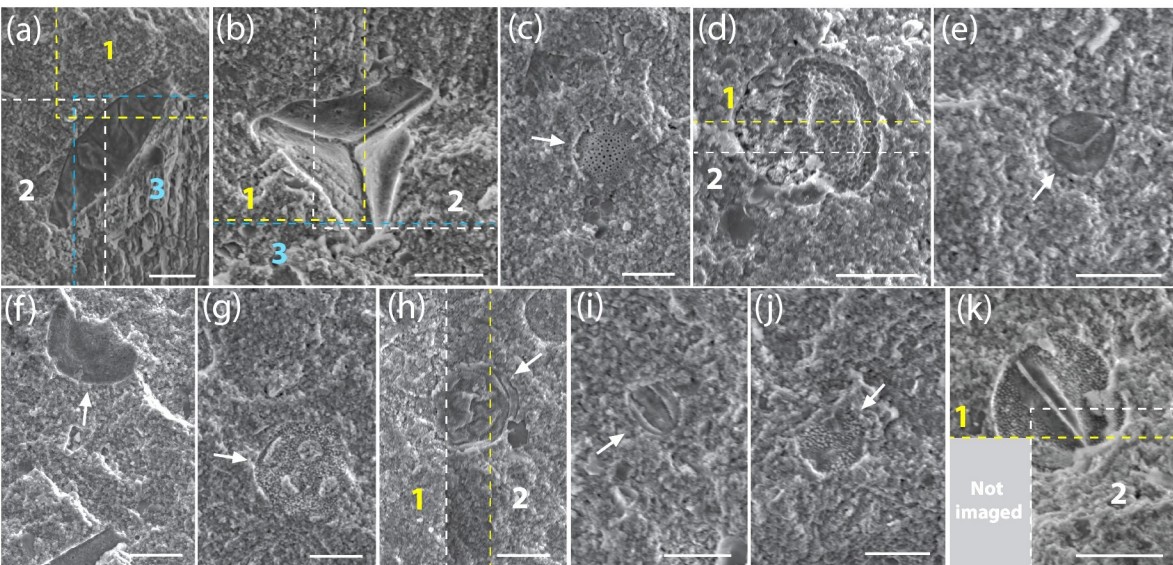

**Fig 6. Pollen and spore types identified.** (*a*) Fungal spore. (*b*) Fern spore. (*c*) *Nothofagidites*. (*d*) Gymnospermous saccate pollen. (*e*) Myrtaceae/Cupanieae pollen. (*f*) Triporate pollen. (*g*) Porate angiosperm pollen. (*h*) Araucariaceae, gymnosperm pollen. (*i*) Angiosperm col (por)ate pollen; *Quintinia*. (*j*) Angiosperm colporate pollen; *Margocolporites vanwijhei*. (*k*) Angiosperm colpate pollen. Dashed lines indicate the overlap/stitching between neighbouring images. Scale bars, 15 μm. Arrows indicate the position of microfossils. Images have been adjusted slightly to improve contrast. Images appearing as they were viewed by volunteers can be found in S3 Fig.

## Accuracy and improvements in citizen scientist microfossil identification

Our approach using a multi-staged identification system (citizen science identification and expert review) was successful in accurately capturing most, if not all, pollen and spores in the large, 25,200-image dataset. The expert review of all 1000 images in Expedition No. 1, revealed that only a few (i.e., seven) indeterminable 'possible' pollen and spores were overlooked by citizen scientists.

A frequent lack of agreement among citizen scientists concerning whether or not an image exhibited pollen or spore/s suggests that there is room for improvement in our approach. Approximately 70–75% of pollen and spores were only identified through the second tier of citizen scientist assessment (i.e., by evaluating 'disputed images') or by expert review of images that had been labelled with 'no agreement' (Fig 3). Upon expert review, however, only a small proportion of images (18% or 289 out of 1588 images) for which 'no agreement', among three citizen scientists, was reached were found to contain pollen or spores. Likewise, upon expert review of 'disputed' images, in which a fourth citizen scientist disputed the 'agreement' of three citizen scientists, only 2% (or 62 out of 2473 images) were found to contain pollen or spores. These shortcomings can be addressed easily by changing the questionnaire. The removal of unnecessary questions (e.g., the fossil position or the number of fossils; Fig 4) is likely to increase the level of agreement among citizen scientists. Providing better citizen scientist training (e.g., through online workshops or more comprehensive teaching material) may further improve accuracy in name identification. These findings are in line with other citizen science projects that analysed a large number of SEM images [8].

## Implications for palaeontology research

Microfossils provide valuable information about the age, fossilisation processes, and palaeoenvironmental characteristics of fossil sites [10, 20–22]. Using the structured approach presented

in this study, citizen scientists successfully helped to identify over 300 fossil pollen and spore specimens from 25,200 SEM images. The automated acquisition of these images enabled us to determine the relative microfossil abundance and palynomorph density/cm$^2$ (Fig 5) and allowed for better use of instrument time. The SEM could be operated overnight and on weekends. Compared to professionals who employ a less structured approach, image review by multiple citizen scientists promoted the discovery of unusual specimens. The microfossils identified here include specimens that are rare or difficult to identify, such as Araucariaceae (gymnosperm) pollen and col(por)ate angiosperm pollen (*Quintinia*) (Fig 6*h* and 6*i*).

These results suggest that our approach could be applied to any fossiliferous rocks that require in situ investigation. For example, petrographic or SEM image analysis for microfossils in silicified fossil deposits [23–25] or processing of aerial photographs of fossil trackways [26] is possible with our approach, given that citizen scientists are guided and that automated image acquisition technology is available.

It is hoped that studies such as ours may stimulate the development of more advanced microscope and imaging software technology, e.g., improved auto-focusing of non-polished and thus uneven rock surfaces, higher image acquisition speed, better image resolution and more user-friendly data managing software. Large datasets could also be processed using machine learning. Such an approach has already been implemented to locate and classify fossil pollen [27], extant pollen [28], and to reconstruct palaeoclimates using stomatal density [28, 29]. A downside to a machine learning approach is that it removes the opportunity for palaeontologists to engage the public in scientific practice. However, validated microfossil image databases are needed to train machine learning algorithms [28, 30]. Therefore, a coupled workflow involving citizen science and machine learning could benefit palaeontological research *and* recruit and train citizen scientists [31].

## Conclusion

The 'Date a Fossil' project successfully engaged 271 citizen scientists to efficiently identify pollen and spore fossils from 25,200 SEM images of the surface of McGraths Flat rocks. Coupled with automated imaging technology, engagement with citizen scientists has proven extremely beneficial. Citizen scientists reduced the large image dataset to 4192 images from which researchers verified 300 pollen and spore fossils. The age and environmental characteristics of this fossil site could be determined using this palynological data. We hope that our results inspire other researchers to consider engaging with citizen scientists to process large image datasets using similar approaches. These methods could be applied to any research that seeks to process a variety of large in situ image datasets while engaging and training the public in palaeontology.

## Supporting information

**S1 Fig. Screenshots of the questionnaire template for the 'Date a Fossil Project' hosted on DigiVol.**
(PDF)

**S2 Fig. Training manual and microfossil identification guide for citizen scientists engaging with 'Date a Fossil' on DigiVol.** Pollen images from page 5 (Fungal spore, Nothofagus pollen), page 8 (Myrtaceae and cupanieidite pollen), and page 10 (angiosperm pollen) are modified from McCurry et al., 2022.
(PDF)

**S3 Fig.** Edited (a-k) and unedited (a-k) micrographs of pollen and spore examples from McGraths Flat. At the centre is Fig 6 from the manuscript, which is surrounded by outset images of the corresponding original unedited SEM micrographs that were viewed by citizen scientists. (a) Fungal spore. (b) Fern spore. (c) *Nothofagidites*. (d) Gymnospermous saccate pollen. (e) Myrtaceae/Cupanieae pollen. (f) Triporate pollen. (g) Porate angiosperm pollen. (h) Araucariaceae, Gymnosperm pollen. (i) Angiosperm col(por)ate pollen; *Quintinia*. (i) Angiosperm colporate pollen; *Margocolporites vanwijhei*. (k) Angiosperm colpate pollen. Dashed lines indicate the overlap/stitching between images. Arrows indicate the microfossils. Scale bars, 15 μm. Field of view for all outset unedited images, 120 μm.
(PDF)

**S1 Data. All raw citizen science expedition data (CSV files) and collated data.**
(ZIP)

**S1 Table. Breakdown of total number of images transcribed by volunteers and expert reviewed.** Total number of images in which volunteers reached agreement or did not reach agreement on the questionnaire template. Total number of images that needed expert review, as well as images verified to contain pollen/spores and final pollen/spore counts.
(PDF)

**S2 Table. Total pollen and spore counts (expert verified) from 25,200 SEM images analysed by citizen scientists and palynomorph density/cm$^2$.**
(PDF)

# Acknowledgments

We are grateful to the Etheridge family descendants, the Australian Museum Research Institute, and the Australian Museum Trust for funding support. Nigel McGrath kindly provided access to the fossil site. We thank the reviewers of this manuscript, who provided valuable feedback. We acknowledge the traditional custodians of the land and waterways on which the McGraths Flat deposit is located, the Wiradjuri Nation People. Lastly, this work would not have been possible without the valuable contributions of volunteer citizen scientists, we recognise and greatly appreciate their time and effort.

# Author Contributions

**Conceptualization:** Tara Djokic, Michael Frese, Matthew R. McCurry.

**Data curation:** Tara Djokic, Adam Woods, Paul Flemons.

**Formal analysis:** Tara Djokic, Michael Frese, Adam Woods, Mary Dettmann, Matthew R. McCurry.

**Funding acquisition:** Michael Frese, Matthew R. McCurry.

**Investigation:** Tara Djokic, Michael Frese, Mary Dettmann, Matthew R. McCurry.

**Methodology:** Tara Djokic, Michael Frese, Adam Woods, Paul Flemons, Matthew R. McCurry.

**Project administration:** Tara Djokic, Adam Woods, Paul Flemons.

**Resources:** Frank Brink.

**Software:** Paul Flemons, Frank Brink.

**Supervision:** Matthew R. McCurry.

**Visualization:** Tara Djokic, Michael Frese, Frank Brink.

**Writing – original draft:** Tara Djokic, Michael Frese, Matthew R. McCurry.

**Writing – review & editing:** Tara Djokic, Michael Frese, Adam Woods, Mary Dettmann, Paul Flemons, Frank Brink, Matthew R. McCurry.

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
