## [Decision Letter · Decision Letter 0]

13 Feb 2023

PONE-D-23-02267Inferring the age and environmental characteristics of fossil sites using citizen sciencePLOS ONE

Dear Dr. Djokic,

Thank you for submitting your manuscript to PLOS ONE. After careful consideration, we feel that it has merit but does not fully meet PLOS ONE’s publication criteria as it currently stands. Therefore, we invite you to submit a revised version of the manuscript that addresses the points raised during the review process.

We look forward to receiving your revised manuscript.

Kind regards,

Huasheng Huang

Academic Editor

PLOS ONE

Journal Requirements:

3. We note that Figures 1, 2, S1 and S2 in your submission contain copyrighted images. All PLOS content is published under the Creative Commons Attribution License (CC BY 4.0), which means that the manuscript, images, and Supporting Information files will be freely available online, and any third party is permitted to access, download, copy, distribute, and use these materials in any way, even commercially, with proper attribution. For more information, see our copyright guidelines: http://journals.plos.org/plosone/s/licenses-and-copyright.

a. You may seek permission from the original copyright holder of Figuresn 1, 2, S1 and S2 to publish the content specifically under the CC BY 4.0 license. 

4. We note that Figure 1 your submission contain map/satellite image which may be copyrighted. All PLOS content is published under the Creative Commons Attribution License (CC BY 4.0), which means that the manuscript, images, and Supporting Information files will be freely available online, and any third party is permitted to access, download, copy, distribute, and use these materials in any way, even commercially, with proper attribution. For these reasons, we cannot publish previously copyrighted maps or satellite images created using proprietary data, such as Google software (Google Maps, Street View, and Earth). For more information, see our copyright guidelines: http://journals.plos.org/plosone/s/licenses-and-copyright.

Additional Editor Comments:

Your manuscript received very good comments from the two reviewers. Please revise based on their comments and suggestions before the publication.

Reviewers' comments:

Reviewer's Responses to Questions

**Comments to the Author**

1. Is the manuscript technically sound, and do the data support the conclusions?

Reviewer #1: Yes

Reviewer #2: Yes

2. Has the statistical analysis been performed appropriately and rigorously? 

Reviewer #1: Yes

Reviewer #2: Yes

3. Have the authors made all data underlying the findings in their manuscript fully available?

Reviewer #1: Yes

Reviewer #2: Yes

4. Is the manuscript presented in an intelligible fashion and written in standard English?

Reviewer #1: Yes

Reviewer #2: Yes

5. Review Comments to the Author

Reviewer #1: This is a very nice paper with an interesting new approach to microfossil identifications using the help of citizen scientists, increasing the identified number of identified microfossils as well as reducing the time of the expert reviewing the material and increasing the accuracy of the fossil identifications.

As observations:

1. It is necessary for the authors to review the non-acid protocol for pollen extraction (Pound et al 2020) and see if this may work to extract the microfossils from iron-oxyhydroxide rocks. This will not change the premise of the paper but will complement and strengthen the methodology presented, especially if the method works for extracting microfossils in rocks that are chemically homogeneous.

In the discussion, the authors mentioned machine learning for the identification of pollen but don’t cite Romero et al 2020 “Improving the taxonomy of fossil pollen using CNN and superresolution microscopy”.

Please review the resolution of the figures because all are pixelated. These need to be at least 300 DPI

Other small comments are in the file attached

Reviewer #2: Dear authors,

I will go straight forward, this manuscript is for me excellent.

I enjoy research that involves citizen science. I believe citizen science is an untapped potential that can be used to do great science and aven make new discoveries.

The large amount of data that has been collected and analyzed is impressive.

Critiquing your own approach is a great way to develop and suggest improvements, especially for this type of approach that you did for this research.

Bravo.

I will have 3 minor comments to suggest.

1/ Figure 1: I would suggest to at least add Canberra on the map (because capital of the country). As you decided to have an "administrative" representation of Australia (with the states) then you should at least add the capital. Also it help people who do not know well the geography of this country, to better situated the location based on city.

2/ Introduction: The problem you propose to resolve is not clearly explained. In the last part of the introduction, we understand that it is intended to "address" the disadvantages mentioned above, but it looks to be presented as the solution already, while we are in the introduction part. To better introduce the problem, it should be rephrased to suggest that a solution using "citizen science" can be used to be compare with the "disadvantages" mentioned above and then discuss what’s the best or at least how “citizen scientist” can help.

3/ The best conclusion of your paper is not the scientific findings themselves (which are certainly important), but the fact that using citizen scientists is beneficial for this type of study. I suggest emphasizing this point in the conclusion, as it currently reads like an abstract. The conclusion could be more engaging and fulfill the "hopes" expressed in the last part of the discussion.

Voilà !

Again, congratulation for this work and initiative.

6. PLOS authors have the option to publish the peer review history of their article (what does this mean?). If published, this will include your full peer review and any attached files.

Reviewer #1: No

Reviewer #2: **Yes: **Benjamin Adroit

---

## [Author Response · Author response to Decision Letter 0]

24 Mar 2023

Manuscript: Inferring the age and environmental characteristics of fossil sites using citizen science

ID: PONE-D-23-02267

We would like to thank the reviewers and editorial team for their consideration of the manuscript. We have listed the responses to the reviewers’ comments below in red text.

Reviewer #1: 

This is a very nice paper with an interesting new approach to microfossil identifications using the help of citizen scientists, increasing the identified number of identified microfossils as well as reducing the time of the expert reviewing the material and increasing the accuracy of the fossil identifications.

As observations:

1. It is necessary for the authors to review the non-acid protocol for pollen extraction (Pound et al 2020) and see if this may work to extract the microfossils from iron-oxyhydroxide rocks. This will not change the premise of the paper but will complement and strengthen the methodology presented, especially if the method works for extracting microfossils in rocks that are chemically homogeneous.

We thank the reviewer for pointing us to these references. We have reviewed their content and expanded the manuscript as suggested.

2. In the discussion, the authors mentioned machine learning for the identification of pollen but don’t cite Romero et al 2020 “Improving the taxonomy of fossil pollen using CNN and superresolution microscopy”.

We have referenced this work and cited it in our text as suggested.

3. Please review the resolution of the figures because all are pixelated. These need to be at least 300 DPI

In the revised manuscript, all images have a resolution of 300 DPI.

4. Other small comments are in the file attached.

Please see below our responses to the in-text comments.

Response to reviewer 1 comments, line-by-line:

Line 56: There are two recent papers about microfossil preparations, one of these is a non-acid protocol. Please review these papers.

Riding_2021_A_guide_to_preparation_protocols_in_palynology

Pound_et_al_2020_overview_of_techniques_applied_to_the_extraction_non-pollen_palynomorphs_taphonomic_issues_recovery

We now cite Riding (2021) and Pound et al (2021).

Lines 209-212: This section needs to be in the results. 

The caption of Fig 3 has been moved to the results section. 

Lines 393-395: This has been also done to classify fossil pollen

Romero, kong et al. 2020 pollen confocal neural network

We have included and cited this work in the revised manuscript.

Reviewer #2: 

Dear authors,

I will go straight forward, this manuscript is for me excellent.

I enjoy research that involves citizen science. I believe citizen science is an untapped potential that can be used to do great science and even make new discoveries.

The large amount of data that has been collected and analyzed is impressive.

Critiquing your own approach is a great way to develop and suggest improvements, especially for this type of approach that you did for this research.

Bravo.

I will have 3 minor comments to suggest.

1/ Figure 1: I would suggest to at least add Canberra on the map (because capital of the country). As you decided to have an "administrative" representation of Australia (with the states) then you should at least add the capital. Also it helps people who do not know well the geography of this country, to better situated the location based on city.

We have added the abbreviations for each Australian state and territory (e.g., NSW, SA, WA, etc) to provide more geographical context for the reader. 

2/ Introduction: The problem you propose to resolve is not clearly explained. In the last part of the introduction, we understand that it is intended to "address" the disadvantages mentioned above, but it looks to be presented as the solution already, while we are in the introduction part. To better introduce the problem, it should be rephrased to suggest that a solution using "citizen science" can be used to be compared with the "disadvantages" mentioned above and then discuss what’s the best or at least how “citizen scientist” can help.

We have revised the introduction as suggested by the reviewer.

3/ The best conclusion of your paper is not the scientific findings themselves (which are certainly important), but the fact that using citizen scientists is beneficial for this type of study. I suggest emphasizing this point in the conclusion, as it currently reads like an abstract. The conclusion could be more engaging and fulfill the "hopes" expressed in the last part of the discussion.

We have revised the conclusion as suggested. 

Voilà !

Again, congratulation for this work and initiative.

Finally, we made additional (minor) changes throughout the manuscript to correct typos, increase clarity, and remove redundancies.

Journal requirements addressed:

1. Style requirements and file naming 

• The manuscript has been formatted as per the style template provided.

• Figure files have been named Fig1.tif, Fig2.tif etc. 

• Supplementary Figure files have been named S1_Fig.PDF, S1_Table.PDF etc.

2. Ethics statement 

Your ethics statement should only appear in the Methods section of your manuscript. If your ethics statement is written in any section besides the Methods, please move it to the Methods section and delete it from any other section. Please ensure that your ethics statement is included in your manuscript, as the ethics statement entered into the online submission form will not be published alongside your manuscript. 

The ethics statement has been moved to the methods section.

3. Figures 1, 2, S1 and S2 

We note that Figures 1, 2, S1 and S2 in your submission contain copyrighted images. All PLOS content is published under the Creative Commons Attribution License (CC BY 4.0), which means that the manuscript, images, and Supporting Information files will be freely available online, and any third party is permitted to access, download, copy, distribute, and use these materials in any way, even commercially, with proper attribution. For more information, see our copyright guidelines: http://journals.plos.org/plosone/s/licenses-and-copyright.

All images used in the figures are either created by the authors, used under a CC BY 4.0 licence, or have been listed with attribution. See the list below for more details:

Fig 1b (backhoe), photo taken by authors.

Fig 1c (ferricrete layers), photo taken by authors.

Fig 1d (leaf), modified from McCurry et al. 2022 using CC BY 4.0 licence.

Fig 1e (flowers), modified from McCurry et al. 2022 using CC BY 4.0 licence.

Fig 1f (sawfly), modified from McCurry et al. 2022 using CC BY 4.0 licence.

• For Figs 1d-f, we have added this reference to the caption text. 

Fig 2a (picture of the SEM), photo taken by authors.

Fig 2b, photo taken by authors.

Fig 2d-e, photos taken by authors.

S1 Fig, all images of pollen (and screenshots) were taken by the authors; the DigiVol website was created and is run by the authors at the Australian Museum. The authors own all rights to use these images. 

S2 Fig (Pollen ID manual), this manual was created by the authors. Images on p. 1-4, 6, 7, 9, 11-13 were taken by the authors. The following images from McCurry et al., Science Advances 2022 have now been cited in S2 Fig:

o p. 5, a fungal spore and two Nothofagidites 

o p. 8, myrtaceous pollen

o p. 10, angiosperm pollen 

The below paper is published under the CC BY 4.0 licence: McCurry MR, Cantrill DJ, Smith PM, Beattie R, Dettmann M, Baranov V, et al. A Lagerstätte from Australia provides insight into the nature of Miocene mesic ecosystems. Science advances. 2022;8(1):e1406. 

4. Map image in Figure 1 

We note that Figure 1 in your submission contains a map/satellite image which may be copyrighted. All PLOS content is published under the Creative Commons Attribution License (CC BY 4.0), which means that the manuscript, images, and Supporting Information files will be freely available online, and any third party is permitted to access, download, copy, distribute, and use these materials in any way, even commercially, with proper attribution. For these reasons, we cannot publish previously copyrighted maps or satellite images created using proprietary data, such as Google software (Google Maps, Street View, and Earth). For more information, see our copyright guidelines: http://journals.plos.org/plosone/s/licenses-and-copyright.

Fig 1a (map) was created by the authors using adobe illustrator. 

5. Review your reference list

Please review your reference list to ensure that it is complete and correct. 

If you have cited papers that have been retracted, please include the rationale for doing so in the manuscript text, or remove these references and replace them with relevant current references. 

Any changes to the reference list should be mentioned in the rebuttal letter that accompanies your revised manuscript. 

If you need to cite a retracted article, indicate the article’s retracted status in the References list and also include a citation and full reference for the retraction notice.

We have reviewed our reference list to check it is complete and correct.

---

## [Editor Report · Decision Letter 1]

29 Mar 2023

Inferring the age and environmental characteristics of fossil sites using citizen science

PONE-D-23-02267R1

Dear Dr. Djokic,

We’re pleased to inform you that your manuscript has been judged scientifically suitable for publication and will be formally accepted for publication once it meets all outstanding technical requirements.

Kind regards,

Huasheng Huang

Academic Editor

PLOS ONE

Additional Editor Comments (optional):

You have greatly improved the manuscript by addressing the reviewers' concerns. Congratulations!
---

## [Editor Report · Acceptance letter]

10 Apr 2023

PONE-D-23-02267R1 

Inferring the age and environmental
characteristics of fossil sites using citizen science 

Dear Dr. Djokic:

I'm pleased to inform you that your manuscript has been deemed suitable for publication in PLOS ONE. Congratulations! Your manuscript is now with our production department. 

Kind regards, 

on behalf of

Dr. Huasheng Huang 

Academic Editor

PLOS ONE